# High-resolution population estimation using household survey data and building footprints

Gianluca Boo [1✉], Edith Darin [1], Douglas R. Leasure[1], Claire A. Dooley [1], Heather R. Chamberlain [1], Attila N. Lázár [1], Kevin Tschirhart[2], Cyrus Sinai [3,4], Nicole A. Hoff [3], Trevon Fuller [3], Kamy Musene[3], Arly Batumbo[5], Anne W. Rimoin [3] & Andrew J. Tatem [1]

The national census is an essential data source to support decision-making in many areas of public interest. However, this data may become outdated during the intercensal period, which can stretch up to several decades. In this study, we develop a Bayesian hierarchical model leveraging recent household surveys and building footprints to produce up-to-date population estimates. We estimate population totals and age and sex breakdowns with associated uncertainty measures within grid cells of approximately 100 m in five provinces of the Democratic Republic of the Congo, a country where the last census was completed in 1984. The model exhibits a very good fit, with an $R^2$ value of 0.79 for out-of-sample predictions of population totals at the microcensus-cluster level and 1.00 for age and sex proportions at the province level. This work confirms the benefits of combining household surveys and building footprints for high-resolution population estimation in countries with outdated censuses.

[1] WorldPop, School of Geography and Environmental Science, University of Southampton, Southampton, UK. [2] Center for International Earth Science Information Network (CIESIN), Columbia University, New York, NY, USA. [3] UCLA Fielding School of Public Health, University of California at Los Angeles, Los Angeles, CA, USA. [4] Department of Geography, University of North Carolina at Chapel Hill, Chapel Hill, NC, USA. [5] Bureau Central du Recensement, Institut National de la Statistique, Kinshasa, Democratic Republic of the Congo. ✉email: gianluca.boo@gmail.com

Accurate population figures are essential to support decision-making in many areas of public interest, for instance, urban planning, environmental hazard risk management, and public health[1]. To this end, the most complete and reliable data source is arguably the national population and housing census[1,2]. However, the data collected in the census may quickly become outdated because of migration, fertility, and mortality patterns occurring during the 10-year intercensal period, which can occasionally stretch up to several decades[3,4]. In such circumstances, outdated census data can be completed using different population estimation techniques. For instance, the Population Division of the United Nations Department of Economic and Social Affairs (UN DESA) produces annual national population estimates using projection models[3]. However, these estimates can be highly uncertain because the models do not address population dynamics occurring at the subnational level, which may fluctuate significantly when the census data is particularly outdated[3,4].

The bottom-up modeling approach addresses the limitations of projection models because it produces population estimates at high spatial resolution independently from the national census[4]. Bottom-up models leverage population data retrieved from recent household surveys involving the complete enumeration of a representative sample of small and well-defined areas, named microcensus clusters[4]. In their basic form, these models link microcensus-cluster-level population totals and ancillary geospatial covariates with complete coverage of the region of interest, such as settlement extents[5,6] and satellite imagery classes[7,8], to estimate population totals in unsurveyed areas. These models can also include additional geospatial covariates[5–8], administrative or functional strata[5,6], and existing age and sex structures to disaggregate the population estimates within different age and sex groups[9]. The United Nations Population Fund (UNFPA) recently highlighted the potential role of bottom-up population models for census planning and preparation[10].

In this work, we develop a Bayesian hierarchical model for bottom-up population estimation that leverages household surveys and building footprints. Our model introduces a weighted-precision approach to derive unbiased estimates of population counts from household surveys with complex sampling designs and modeled age and sex structures using the same survey data. It also makes extensive use of building footprints to approximate the extent and distribution of settled areas and derive morphological and topological attributes implemented as geospatial covariates and in a functional classification of settlement types. We model population totals and age and sex breakdowns with associated uncertainty measures within grid cells of approximately 100 m in five provinces in the western part of the Democratic Republic of the Congo (DRC). Our estimates are publicly available and can be flexibly aggregated within different geographic units to support decision-making in a country where uncertainty around the geography and demography of its population regularly hampers public health and humanitarian interventions.

## Results

**Population estimates.** We developed a hierarchical Bayesian model to estimate population totals and age and sex breakdowns at high spatial resolution in five provinces in the western part of the DRC. Figure 1 shows the estimated population totals within grid cells of approximately 100 m for the five provincial capitals. Kinshasa (Fig. 1a) has the largest spatial extent and the highest population totals per grid cell, with substantial variations between the central part of the city and its outskirts. The remaining cities (Fig. 1b-e) have reduced extents and lower population totals per grid cell, confirming the predominantly rural character of the study region. The high-resolution population estimates, including totals, age and sex breakdowns, and associated measures of model uncertainty, can be accessed on the WorldPop Open Population Repository (WOPR)[11], visualized using the WOPR Vision web application[12], and integrated into data analyses through the WOPR R package[13]. These tools can also facilitate the localized comparison of the high-resolution population estimates with the enumeration of small geographic areas frequently carried out as part of public health campaigns[14,15].

**Population totals and densities.** As shown in Eq. (1), we modeled population totals as a Poisson process resulting from estimated population densities multiplied by the total area of building footprints within the microcensus clusters. We computed population totals across 926 clusters based on data collected across two rounds of household surveys and the total area of the building footprints. In doing so, we discarded 21 clusters exhibiting spurious attributes: 7 clusters featured substantial undercounting of people associated with limited survey coverage, while in 14 clusters no building footprints were detected. Figure 2 shows the geographic distribution of the observed population densities across the 905 microcensus clusters according to the settlement type and province. The clusters in the provinces of Kwango (Fig. 2c), Kwilu (Fig. 2d), Mai-Ndombe (Fig. 2e), the eastern part of Kinshasa (Fig. 2a), and Kongo Central (Fig. 2b) were primarily rural, with highly heterogeneous population densities. Most urban clusters were located in the provinces of Kinshasa and Kongo Central, the most urbanized parts of the study region, where population densities were generally homogeneous and lower than in rural clusters.

**Hierarchical intercepts.** As presented in Eq. (2), we defined population log-densities in the 905 microcensus clusters as the response variable of a linear regression. We estimated the random intercept hierarchically by settlement type ($n = 2$), province ($n = 5$), and sub-provincial region comprising territories, cities, and groups of municipalities for the city-province of Kinshasa ($n = 37$) (Eq. (8)). Figure 3 shows the posterior probability distributions of the hierarchical intercept by settlement type and province. While the distributions were generally similar across rural and urban settlements, the posterior means were lower in the urban settlements of the Kinshasa (Fig. 3a) and Kongo Central (Fig. 3b) provinces, potentially because of the higher prevalence of building footprints with a non-residential function. The 95% credible intervals are wider in urban settlements, especially in the provinces of Kongo Central (Fig. 3b), Kwango (Fig. 3c), and particularly Mai-Ndombe (Fig. 3e), most likely because of the large diversity of urban settlements, ranging from sparsely populated towns and suburban areas to denser city centers.

**Covariate effects.** As shown in Eq. (7), the hierarchical intercepts presented above were combined with the additive effects of three geospatial covariates derived from building footprint attributes. In doing so, we first defined the covariate effects to be independently estimated by settlement type for each covariate (Eq. (9)) and, if their posterior distribution was similar across rural and urban settlements, we converted them into fixed effects (Eq. (10)). Figure 4 shows the covariate effects estimated in the model; we estimated random effects by settlement type for the first two covariates and a fixed effect for the third covariate. While the covariate average building proximity (i.e., the inverse of the average distance to the nearest building footprint) had a significant positive effect at the 95% credible level in rural

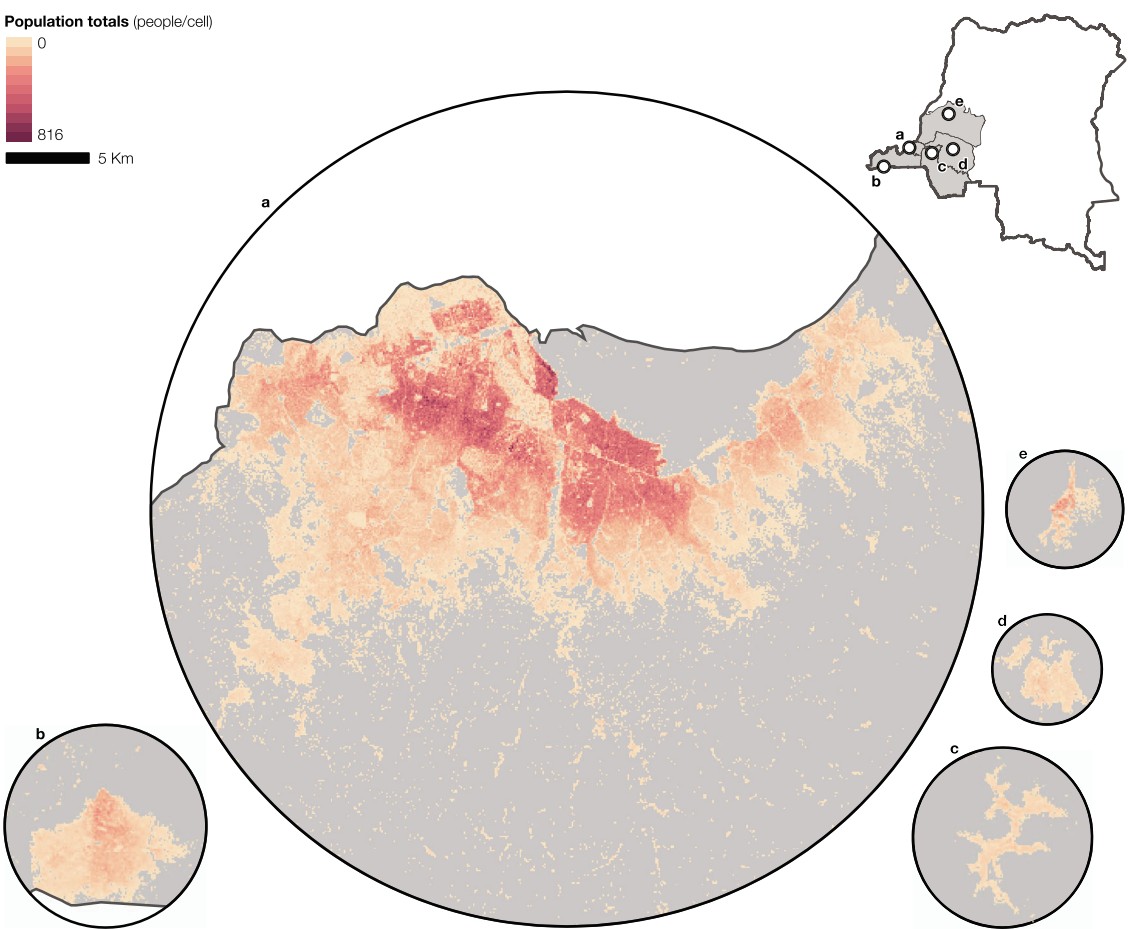

**Fig. 1 Gridded population estimates in selected cities.** Estimated population totals (people/cell) in the capital cities of the provinces of (**a**) Kinshasa (Kinshasa), (**b**) Kongo Central (Matadi), (**c**) Kwango (Kenge), (**d**) Kwilu (Bandundu), and (**e**) Mai-Ndombe (Inongo). The estimates represent the mean of the posterior distribution ($n = 10,000$). The map of the DRC shows the extent of the five provinces defining the study region in gray.

settlements, the effect was non-significant in urban settlements. Conversely, the covariate average building focal count (i.e., the average count of building footprints in a focal window of approximately 2 km) had a significant negative effect at the 95% credible level in rural settlements and a significant positive effect in urban settlements. The covariate average building area (i.e., the average area of the building footprints) had a strong significant negative effect at the 95% level across both settlement types and was converted into a fixed effect.

**Age and sex proportions**. As presented in Eqs. (11, 12), we modeled age and sex proportions from the household survey data aggregated at the province level as a Dirichlet-multinomial process. Figure 5 shows the means of the posterior distribution of the age and sex proportions with relative 95% credible intervals for the five provinces. Age and sex structures were similar in the predominantly rural provinces of Kwango (Fig. 5c), Kwilu (Fig. 5d), and Mai-Ndombe (Fig. 5e). In these provinces, the bases of the pyramids were large and became increasingly narrow for older age groups. In the predominantly urban provinces of Kinshasa (Fig. 5a) and Kongo Central (Fig. 5b), the pyramids had a narrower base, typically associated with lower fertility. The province of Kinshasa also had a larger part of the population between 20 and 49 years old because of work-related migratory patterns. The 95% credible intervals were generally narrow as a consequence of the limited variation in the aggregated province-level age and sex structures.

**Model diagnostics**. We achieved model convergence in 10,000 sample iterations for the three Markov chain Monte Carlo (MCMC) chains. Table 1 summarizes the analysis of residuals for population totals (people) and population densities (people/ building footprint ha) at the microcensus-cluster level and age and sex proportions at the cluster level for in-sample and out-of-sample posterior predictions. The analysis suggested a very good model fit for population totals for in-sample ($R^2 = 0.81$) and out-of-sample ($R^2 = 0.79$) predictions, despite a reduced fit for population densities for in-sample ($R^2 = 0.52$) and out-of-sample ($R^2 = 0.47$) predictions. For both population totals and densities, approximately 90% of the observations were within the 95% credible intervals of out-of-sample predictions, suggesting that the uncertainty intervals were robust. The analysis also indicated slight bias with over-prediction of population totals and under-prediction of population densities at the microcensus-cluster level, with larger imprecision and inaccuracy for the latter. Both for population totals and densities, no significant ($p < 0.05$) spatial autocorrelation in the model residuals was detected using the Moran's I test. Province-level age and sex proportions had a perfect fit for in-sample ($R^2 = 1.00$) and out-of-sample ($R^2 = 1.00$) model predictions, with imperceptible levels of imprecision and inaccuracy, suggesting limited variability in the age and sex proportions aggregated at the province level. In addition, in both cases, 100% of the observations fell within the 95% credible intervals, indicating conservative uncertainty intervals for age and sex proportions estimated at the province level.

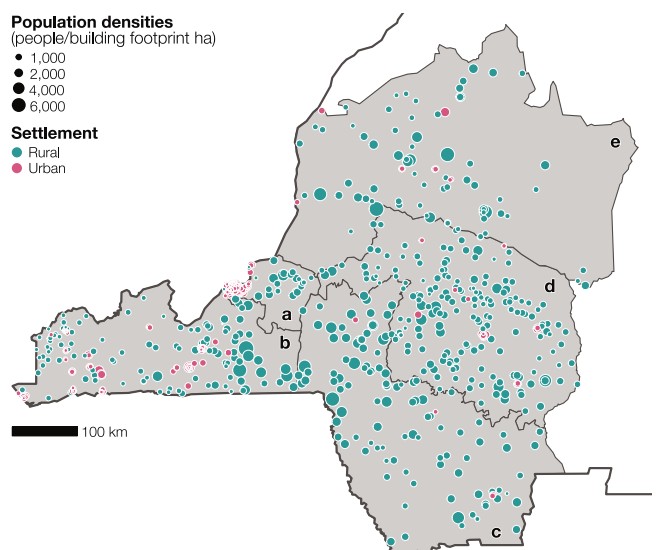

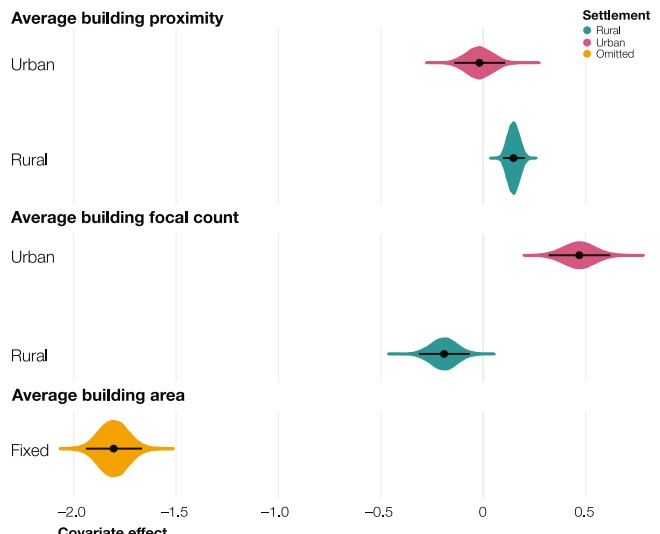

**Fig. 4 Covariates effect.** Posterior probability distribution of the random effect by settlement type (urban in pink and rural in purple) for the covariates average building proximity and average building focal count and the fixed effect (in orange) for the covariate average building area. The black dots show the mean of the distributions and the horizontal black lines show the 95% credible intervals derived from the posterior distribution ($n = 10,000$). Source data are provided with this paper.

**Fig. 2 Microcensus clusters location and associated population densities.** Observed population densities (people/building footprint ha) across the microcensus clusters ($n = 905$ clusters) in the provinces of (**a**) Kinshasa ($n = 26$ rural and $n = 254$ urban clusters), (**b**) Kongo Central ($n = 113$ rural and $n = 109$ urban clusters), (**c**) Kwango ($n = 111$ rural and $n = 6$ urban clusters), (**d**) Kwilu ($n = 182$ rural and $n = 23$ urban clusters), and (**e**) Mai-Ndombe ($n = 69$ rural and $n = 12$ urban clusters). The microcensus clusters are classified according to the settlement type (urban in pink and rural in turquoise).

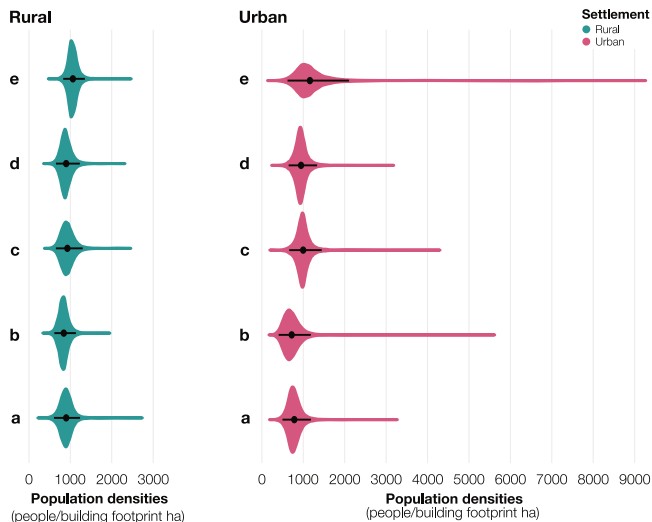

**Fig. 3 Model intercepts.** Posterior probability distributions of the random intercepts (people/building footprint ha) by settlement type (urban in pink and rural cluster in turquoise) across the provinces of (**a**) Kinshasa, (**b**) Kongo Central, (**c**) Kwango, (**d**) Kwilu, and (**e**) Mai-Ndombe. The black dots show the mean of the distributions and the horizontal black lines show the 95% credible intervals derived from the posterior distribution ($n = 10,000$). Source data are provided with this paper.

Figure 6 visually contrasts the observed population totals (people) and densities (people/ building footprint ha) versus in-sample and out-of-sample posterior predictions according to settlement type. The figure confirms a very good model fit for population totals, which were generally in line with the observed

totals. The reduced model fit for population densities appears to be due to large underpredictions in densely populated clusters in rural settlements. In these clusters, underpredictions were related to issues in the building footprint data used to compute the observed population densities because the number of building footprints was substantially lower than the number of surveyed buildings. The reason for this could be found in the satellite imagery used to delineate building footprints that, in these clusters, were partly obfuscated by clouds and smoke, presumably from slash and burn agriculture, that prevented the accurate detection of buildings.

## Discussion

This study extended a Bayesian hierarchical framework for bottom-up population modeling to leverage household survey data with complex sampling designs and building footprint attributes. Existing techniques for population estimation demonstrated the advantages of using household surveys in terms of time and cost to fully enumerate a set of representative clusters compared with the country-wide coverage of the national census[4]. While these estimation techniques are generally endorsed for decision-making[1,2] and census support[10], the resulting estimates are not a substitute for the richness of information collected in the census. However, when the census data is incomplete or outdated, bottom-up models produce comprehensive and up-to-date population estimates at high spatial resolution[4]. The population estimates and associated uncertainty measures can be flexibly aggregated within different units, for instance, administrative boundaries, catchment areas, health districts, and custom-made polygons, to support different applications[16]. These applications can also include validating the modeling results that, in the absence of recent subnational population figures, can be compared with ad-hoc small-area enumerations typically performed as part of public health campaigns[14,15].

Our modeling effort estimated population totals and age and sex breakdowns at a spatial resolution of approximately 100 m

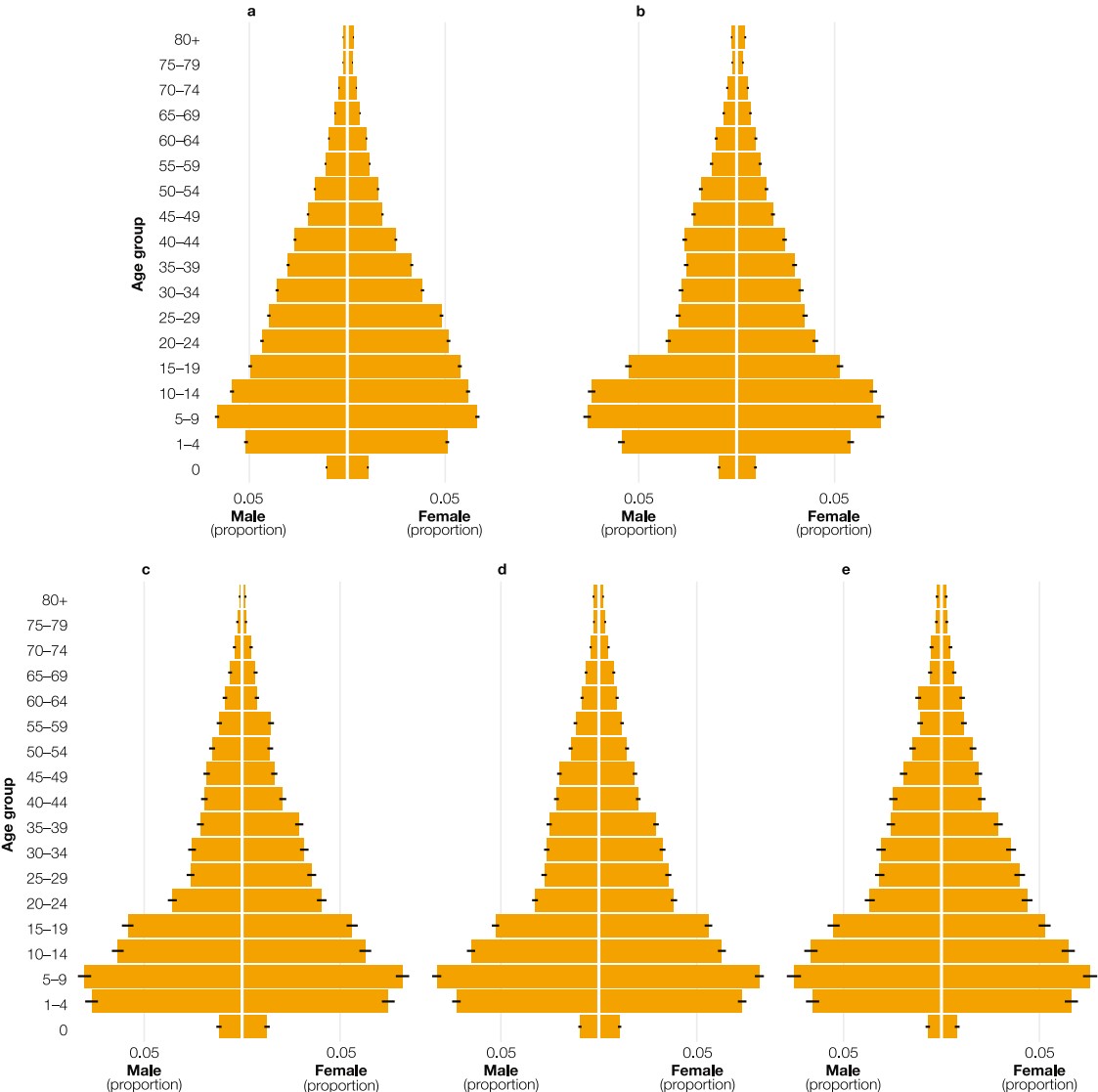

**Fig. 5 Predicted age and sex structures.** Population pyramids presenting the means of the posterior distribution ($n = 10,000$) for each age and sex proportion for the provinces of (**a**) Kinshasa, (**b**) Kongo Central, (**c**) Kwango, (**d**) Kwilu, and (**e**) Mai-Ndombe. The horizontal black lines show the 95% credible intervals derived from the respective posterior distribution ($n = 10,000$). Source data are provided with this paper.

**Table 1 Goodness-of-fit metrics.**

| Estimate | Prediction | Bias | Imprecision | Inaccuracy | $R^2$ | 95% CI |
|---|---|---|---|---|---|---|
| Population totals | In-sample | 13.81 (−0.02) | 165.50 (0.41) | 100.17 (0.27) | 0.81 | 92.49% |
| Population totals | Out-of-sample | 13.43 (−0.03) | 173.28 (0.44) | 105.61 (0.29) | 0.79 | 90.50% |
| Population densities | In-sample | −13.96 (−0.02) | 441.66 (0.41) | 266.32 (0.27) | 0.52 | 92.04% |
| Population densities | Out-of-sample | −15.20 (−0.03) | 464.69 (0.44) | 282.90 (0.29) | 0.47 | 90.06% |
| Age and sex proportions | In-sample | 0.00 (0.00) | 0.00 (0.00) | 0.00 (0.00) | 1.00 | 100.00% |
| Age and sex proportions | Out-of-sample | 0.00 (0.00) | 0.00 (0.00) | 0.00 (0.00) | 1.00 | 100.00% |

Analysis of residuals for the estimated population totals (people), population densities (people/building footprint ha), and province-level age and sex proportions for in-sample and out-of-sample posterior predictions. Bias represents the mean of the residuals, imprecision the standard deviation of residuals, inaccuracy the mean of absolute residuals, $R^2$ the squared Pearson correlation coefficient among the residuals, and the percentage of observations falling within the 95% credible intervals. Values in parentheses are computed using scaled residuals (residuals/predictions). Out-of-sample predictions are carried out using 10-fold cross-validation, where the model is fit ten times, each time withholding a random 10% of microcensus clusters until all is held out once.

together with measures of model uncertainty. The model leveraged household survey data with a probabilistic sampling design[17], typically adopted in national household surveys, such as Demographic and Health Surveys (DHS) and Multiple Indicator Cluster Surveys (MICS)[18]. Given that this type of design oversamples locations with higher population densities, we included a weighted-precision approach to recover unbiased estimates of population totals and densities with robust credible intervals[19]. However, this approach is often difficult to implement[20], as we confirmed by assessing the weights used in

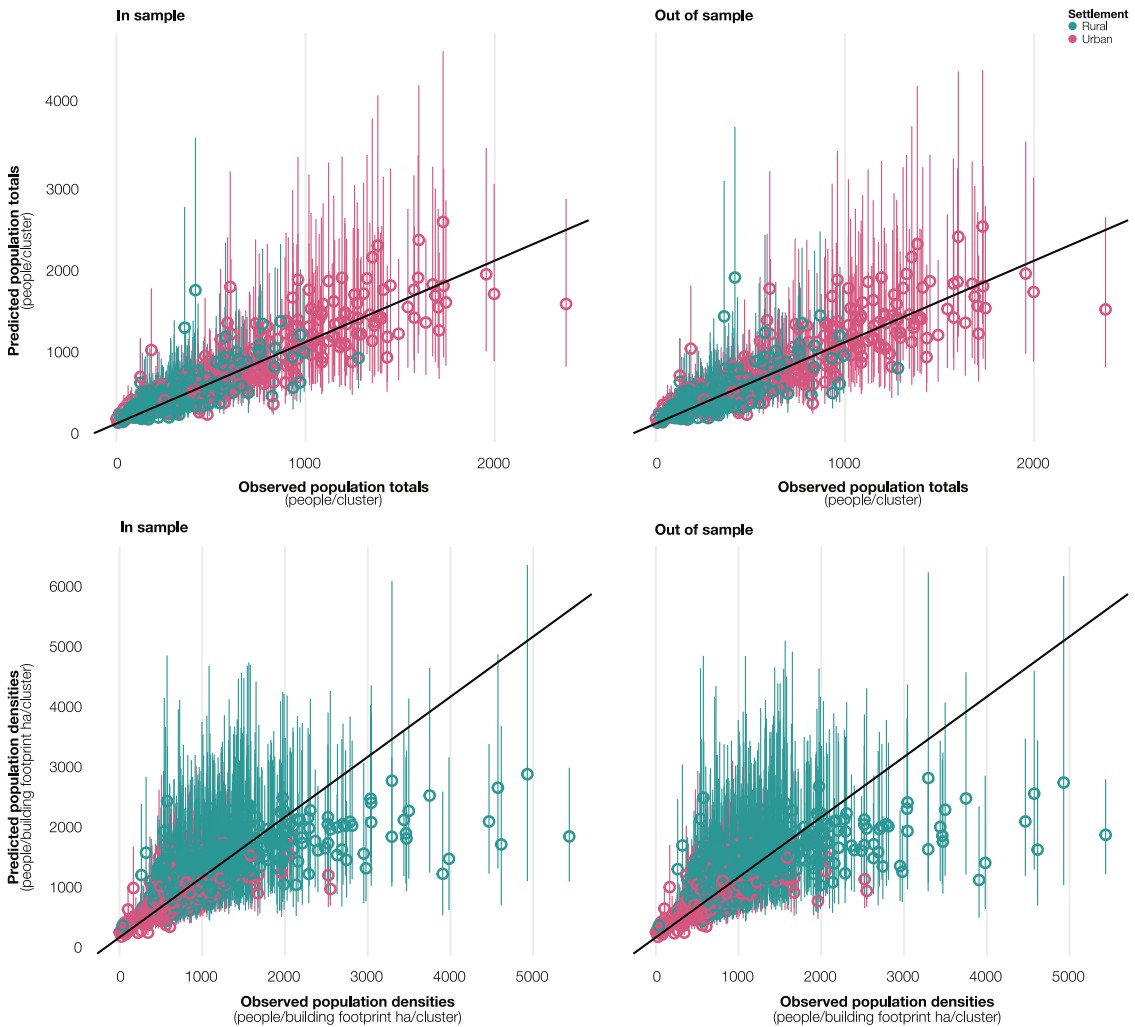

**Fig. 6 Predicted versus observed population totals and densities.** Observed population totals (people/cluster) and densities (people/building footprint ha/cluster) versus in-sample and out-of-sample mean posterior predictions (colored dots) with 95% credible intervals (colored vertical lines) derived from the respective posterior distribution ($n = 10{,}000$). Population totals and densities are classified according to the settlement type (urban in pink and rural in turquoise). The diagonal black lines show a perfect relationship between observations and predictions. Source data are provided with this paper.

the population-weighted sampling. Our assessment identified the presence of weight outliers associated with uncertainties in the population data used to compute the sampling weights that we subsequently truncated at the 90th percentile of the statistical distribution. The use of sampling weights associated with the seeds of the clusters also introduces additional uncertainty because it does not accurately represent the probability of selecting every location in the cluster. We also modeled age and sex proportions at the province level because reduced sample size at finer administrative levels could result in spurious estimates for the smallest groups[9,21]. However, these results should be interpreted with caution because small differences in the estimated proportions (e.g., 0.001) can have a substantial impact on the total population allocated to each class in a province.

We accessed building footprints to approximate the extent and distribution of the settled areas and derive morphological and topological attributes[22–24]. These attributes were used as model covariates and to define hierarchical intercepts and random effects by settlement type. The hierarchical intercept by settlement type and province suggested higher population densities (people/building footprint ha) in rural than urban settlements. The analysis of the observed population densities at the microcensus-cluster level confirmed this counterintuitive result,

where reduced population densities in urban settlements were linked to a higher prevalence of building footprints with non-residential functions (e.g., factories and shops). In addition, we identified issues in the building footprint data associated with outdated input imagery and other contextual factors that prevented the detection of buildings in some clusters. Due to the reduced extent of the observed building footprints, these clusters exhibited inflated observed population densities and consequent substantial model underpredictions. However, in the same clusters, no critical underpredictions were detected for population totals because the areal extent of the observed building footprints acted as a multiplicative constraint in the process of estimation. Considering that our study aimed at estimating population totals, a reduced goodness-of-fit for population densities was considered an endorsement of our hierarchical modeling approach rather than a limitation.

Despite the issues described above, the use of building-footprints attributes as model covariates enabled us to interpret the direction and significance of their associations with population density. The covariate average building proximity had a positive effect on population densities in rural settlements, suggesting a link between population density and settlement compactness. The same covariate had a non-significant effect in urban

settlements, potentially because of the more complex settlement structure[23] and the higher prevalence of building footprints with a non-residential function[23]. The covariate average building focal count had a positive effect on population densities in urban settlements, confirming the association between population density and urban centrality[23,25]. However, we observed an opposite effect in rural settlements, potentially because of higher population densities in settlements with a larger prevalence of building footprints with a prominent residential function[25]. Lastly, the covariate average building area had a consistent negative effect on population densities, suggesting the impact of large building footprints with non-residential function and small but over-crowded building footprints on population densities[23,25]. Given that the covariates presented above were not selected for causal inference, the interpretation of the effects provided above remains hypothetical.

Similar to existing modeling efforts, our model assumed that the population totals retrieved from the household surveys were observed without error and that no people lived outside of the area defined using the building footprints[5,6]. Observation error in the household surveys is likely to result in lower population totals because of inaccessible areas within the microcensus clusters. Additional observation error in the building footprints is expected to underestimate the population totals where the satellite imagery used for automatic extraction was outdated or obfuscated by contextual factors, such as clouds, smoke due to slash-and-burn agriculture, and forest canopy coverage. If these sources of error are systematic, they may be tackled in future studies by including a measurement error component in the model[6]. The model also assumed constant age and sex structures within each province, thus neglecting sub-provincial variations. Whether this choice was dictated by sample size considerations[9], future studies may benefit from a more complex hierarchical structure to capture variations occurring at finer administrative levels[21]. Lastly, future studies will also benefit from a more systematic assessment of the implementation of weighted-precision approaches, particularly in the uncertainty associated with sampling weights[18,19].

This study advances the state-of-the-art of bottom-up population modeling in countries with outdated census data. In particular, our weighted-precision approach allows for the inclusion of household survey data with probabilistic sampling designs (e.g., DHS and MICS), provided that the data includes the sampling weights and precise geographic information on the surveyed clusters, such as the cluster boundary or the GPS location of the household[4]. Our model also estimates age and sex structures using the household survey data, thus providing more up-to-date decompositions than those computed using existing and potentially outdated age and sex structures[9]. Lastly, we also introduce morphological and topological attributes derived from building footprints in bottom-up population modeling. Given that most of these attributes are openly available across sub-Saharan Africa, they could be useful in similar modeling efforts[22]. However, it remains challenging to develop a meaningful comparison between the bottom-up population estimates and official figures because the last national population and housing census of DRC has been completed in 1984, and no other systematic enumeration of the country was carried out since[26,27]. In addition, existing model projections suffer from additional sources of uncertainty associated with questionable assumptions (e.g., use of fragmented or unreliable data[26]) and limitations (e.g., unavailability of sub-national figures[3]) that prevent them from reflecting the spatial distribution of the Congolese population at the subnational level. These assessments can also be affected by uncertainty around the spatial extent of the units of comparison since administrative boundaries often differ according to the data provider. For this reason, aggerated bottom-up estimates may conflict with other population figures because the units of aggregation and enumeration are not the same.

Research to further develop bottom-up population models is underway in the eastern part of the DRC and other countries of sub-Saharan Africa as part of the Geo-Referenced Infrastructure and Demographic Data for Development (GRID3) program[28,29]. These modeling efforts are tailored to the country context and the available input data. Importantly, when the input data is scarce or suboptimal, a limited overall goodness-of-fit of the model does not preclude the successful use of uncertainty measures for policy planning and campaign implementation[14,15]. Although the applications of bottom-up population modeling, conducted as part of the GRID3 program, are primarily focused in sub-Saharan Africa, this approach has broader applicability and can support different steps of the census implementation, from planning (e.g., updating sampling frames) to implementation (e.g., support planning and logistics), from quality assessment (e.g., assess census coverage) to data usage (e.g., data anonymization)[10].

## Methods

**Household surveys.** We accessed geolocated household surveys involving the complete enumeration of 926 microcensus clusters of approximately three settled hectares in five provinces in the western part of the DRC. The data was collected across two rounds of household surveys led by the UCLA-DRC Health Research and Training Program based at the University of California, Los Angeles Fielding School of Public Health and the Kinshasa School of Public Health (KSPH)[30]. The first round of surveys was carried out between May and July 2017 in the provinces of Kinshasa, Kwango, Kwilu, and Mai-Ndombe using random sampling, while the second round was carried out between October and December 2018 in the provinces of Kinshasa and Kongo Central using population-weighted sampling[16]. The surveys were developed for the bottom-up population modeling in the provinces mentioned above. Given the time and resources needed to travel to and fully enumerate the 926 clusters, only essential demographic data (e.g., household size and age and sex characteristics) were collected in the surveys. All survey data were anonymous and purged of any personally identifiable information before being received for analysis.

In both surveys, seed locations (i.e., 100 m grid cells) were first selected, and cluster boundaries were subsequently manually delineated around these locations to include approximately three settled hectares with similar settlement characteristics (e.g., building size and shape) assessed from satellite imagery. We accessed the sampling weights for the second round of surveys and assessed their statistical distribution to identify outliers associated with known uncertainties in the gridded population data used in the sampling[16,18]. To limit the effects of outliers resulting from the gridded population data used in the sampling, we truncated the sampling weights at the 90th percentile of the statistical distribution. We retrieved population totals for the clusters from the population counts recorded within each household where informed consent was obtained ($n = 79{,}126$) and imputed population in households with a nonresponse ($n = 629$) based on the mean population per household within the same cluster. We also retrieved population totals for standardized age (i.e., under 1-year-old, 5-year groups from 1 to 80, and above 80 years old) and sex (i.e., male and female) groups within each province by aggregating individual survey records.

Ethical compliance for the data collection was approved by the institutional review boards at the University of Kinshasa School of Public Health (KSPH) Ethics Committee and at the University of California Los Angeles Institutional Review Board (UCLA IRB). Ethical approval for the data analysis was granted at the University of Southampton Ethics Committee.

**Building footprints.** We accessed building footprint data automatically extracted by Ecopia.AI in 2019 using satellite imagery provided by Maxar Technologies within the DRC[31]. The imagery used for feature extraction provides the best quality (i.e., less than 5% cloud coverage and 0.3% coverage gaps) and the most recent (i.e., on average more recent than 2017) representation of man-made structures visible on the ground, including both residential and non-residential buildings. However, outdated satellite imagery (i.e., dating back to 2009) and other contextual factors (e.g., clouds, smoke due to slash-and-burn agriculture, and forest canopy coverage) may affect the automatic extraction of building footprints in the most dynamic and remote settlement types. Given the robust quality control process developed by Ecopia.AI, the building footprints are considered to provide the most accurate and recent approximation of the spatial distribution of populations across the five provinces. Some alternative building footprint datasets (e.g., Microsoft Building Footprint Data) have recently been released, but their coverage of the DRC is not yet available or optimal.

We used the building footprints to derive morphological and topological attributes, such as area, perimeter, number of nodes, and distance to the nearest feature[22,23]. We summarized these attributes within the microcensus clusters and

grid cells of approximately 100 m comprising the five provinces using basic summary statistics, such as the sum, mean, and coefficient of variation[22]. We also produced the same summary statistics for focal windows of approximately 500 m, 1 km, and 2 km to reflect contextual characteristics[23]. We allocated the microcensus clusters and the grid cells to urban and rural settlements using an existing morphological classification derived from the same building footprint data by the Center for International Earth Science Information Network (CIESIN)[32]. We labeled the original built-up area class as urban settlement and merged the original classes small settlement area (i.e., representing rural settlements) and hamlet (i.e., isolated rural settlements) into a class labeled rural settlement. Urban settlements are characterized by contours with an area greater than or equal to 40 building footprint ha with a building density of at least thirteen building footprints across it, while rural settlements include the remaining part of the study area[32].

**Administrative boundaries**. We accessed administrative boundaries provided by the Bureau Central du Recensement (BCR), the administrative body responsible for the census implementation in the DRC[33]. The boundaries comprised the administrative level 0 (i.e., country), level 1 (i.e., provinces), level 2 (i.e., territories and cities), and level 3 (i.e., sectors/chiefdoms and municipalities). At the time of this study, the administrative boundaries were being consolidated, and level 3 boundaries were only available for the city of Kinshasa. We first derived the spatial extent of the provinces from the level 1 boundaries and subsequently created the sub-provincial regions by combining level 2 and level 3 boundaries. In doing so, we merged the level 3 boundaries of the 24 municipalities comprising the city of Kinshasa into nine contiguous groups of municipalities with similar settlement characteristics as reported in the Strategic Orientation Plan for the Agglomeration of Kinshasa[34]. For instance, the boundaries of the municipalities of Bandalungwa, Kintambo, Ngaliema, and Selembao were merged into a group corresponding to the western expansion of the city. This ad-hoc grouping of municipalities ensured that every sub-provincial region would contain at least one microcensus cluster to estimate random intercepts in the population model. Lastly, we produced gridded datasets with a resolution of approximately 100 m with unique identifiers for each province and sub-provincial region and subsequently allocated the microcensus clusters to a single province and a sub-provincial region.

**Covariate processing and selection**. We first constrained the extent of the clusters using the building footprints located within a radius of approximately 50 m from the surveyed households to exclude areas that were not surveyed because of accessibility constraints. We then derived morphological and topological attribute summaries from the building footprints and extracted additional summaries from standard gridded datasets used in the study of population distributions, for instance, temperature, precipitation, land use, and night-time light intensity[35]. Model covariates were selected by assessing relationships between log-population densities (people/building footprint ha) and the attribute summaries across the clusters using scatterplots and Pearson correlations. This procedure enabled us to retain the five covariates with the strongest linear association to population densities — (1) building count (count of structures), (2) average building area (in ha), (3) average building perimeter (in m), (4) average building proximity or the inverse of the distance to the nearest building (in m), and (5) average building focal count (average count of building within a focal window of approximately 2 km). To avoid multicollinearity, we assessed Pearson correlations between the five covariates and subsequently discarded average building perimeter because it was strongly correlated with average auilding area. Building count was also discarded to avoid potential data circularity because it was used in other parts of the model. The selected covariates were finally scaled based on the mean and standard deviation computed at the grid cell level across the study area.

Data processing and covariate selection were conducted in R version 4.0.2[36] using the R packages raster[37] version 3.0 and sf[38] version 0.7.

**Population model**. We modeled population totals by extending an existing hierarchical Bayesian modeling framework for population estimation[6]. The hierarchical modeling framework offers great flexibility and adaptability to complex input data, such as household survey data, while accurately reflecting model uncertainty through Bayesian credibility intervals. Model uncertainty is associated with the inability to capture features in the input data, for instance, observational error or limited sample size. Equation (1) models the total number of people $N_i$ as a Poisson process, where $D_i$ is the population density (people/building footprint ha) and $A_i$ is the total area of building footprints (ha) derived from the building footprints within each microcensus cluster $i$. The use of building footprints provides a valuable additional source of information that constrains the estimation of population totals within a reasonable range.

$$N_i \sim \text{Poisson}\left(D_i A_i\right) \quad (1)$$

Equation (2) models $D_i$ as a log-normal process to relax the assumptions of the Poisson distribution, where $\bar{D}_i$ is the expected population density on a log-scale and $\tau_{t,p,i}$ is a hierarchical precision term estimated by settlement type $t$ and province $p$ for each cluster $i$.

$$D_i \sim \text{LogNormal}\left(\bar{D}_i, \tau_{t,p,i}\right) \quad (2)$$

Equation (3) defines the precision term $\tau_{t,p,i}$ based on a hierarchical estimate of precision $\tau_{t,p}$ and the model weights $v_i$[19]. $\tau_{t,p}$ is estimated hierarchically by settlement type $t$ and province $p$ using uninformative priors on the mean $\mu_t$ and the variance $\sigma_t$ terms, which are modeled by a normal and uniform distribution, respectively.

$$\tau_{t,p,i} = \sqrt{\frac{1}{v_i \tau_{t,p}^{-2}}} \quad (3)$$

$$\tau_{t,p} \sim \text{Half-Normal}\left(\mu_{t,p}, \sigma_{t,p}\right)$$

$$\mu_{t,p} \sim \text{Half-Normal}\left(\mu_t, \sigma_{t,}\right)$$

$$\sigma_{t,p} \sim \text{Uniform}\left(0, \sigma_t\right)$$

$$\mu_t \sim \text{Normal}\left(0, 1000\right)$$

$$\sigma_t \sim \text{Uniform}\left(0, 1000\right)$$

Equation (4) defines the model weight $v_i$ as the inverse of the sampling weight $w_i$ used to select cluster $i$ in the second round of household surveys. The sum of $w_i$ is used to proportionally impute $w_i$ for the clusters that were selected randomly during the first round of household surveys. $v_i$ are then rescaled to sum to one across all the clusters $I$

$$v_i = \frac{w_i^{-1}}{\sum_{i=1}^{I} w_i^{-1}} \quad (4)$$

As the estimate of precision $\tau_{t,p,i}$ cannot be derived in locations where the model weights $w_i$ are not available and adopted for posterior model predictions, Eq. (5) determines a hierarchical estimate of precision $\hat{\tau}_{t,p}$ from a weighted average of $\tau_{t,p,i}$, where $I_{t,p}$ is the number of clusters $i$ within settlement type $t$ and province $p$.

$$\hat{\tau}_{t,p} = \frac{\sum_{i=1}^{I_{t,p}} \tau_{t,p,i} \sqrt{v_i}}{\sum_{i=1}^{I_{t,p}} \sqrt{v_i}} \quad (5)$$

Equation (6) uses the precision estimate $\hat{\tau}_{t,p}$ for posterior model predictions by altering Eq. (2).

$$\hat{D}_i \sim \text{LogNormal}\left(\bar{D}_i, \hat{\tau}_{t,p}\right) \quad (6)$$

Equation (7) models the expected population density $\bar{D}_i$ using a linear regression with random intercept $\alpha_{t,p,l}$ estimated by settlement type $t$, province $p$, and local area $l$ and $K$ covariates $x_k$ with random effects $\beta_{k,t}$ estimated by settlement type $t$.

$$\bar{D}_i = \alpha_{t,p,l} + \sum_{k=1}^{K} \beta_{k,t} x_{k,i} \quad (7)$$

Equation (8) models the hierarchical intercept $\alpha_{t,p,l}$ for a local area $l$ belonging to a settlement type $t$ and province $p$ as a nested hierarchy with uninformative priors on the mean $\xi_{t,p}$ and variance $\nu_{t,p}$ terms. These are modeled using a normal and uniform distribution, respectively.

$$\alpha_{t,p,l} \sim \text{Normal}\left(\xi_{t,p}, \nu_{t,p}\right) \quad (8)$$

$$\xi_{t,p} \sim \text{Normal}\left(\xi_t, \nu_t\right)$$

$$\nu_{t,p} \sim \text{Uniform}\left(0, \nu_t\right)$$

$$\xi_t \sim \text{Normal}\left(0, 1000\right)$$

$$\nu_t \sim \text{Uniform}\left(0, 1000\right)$$

Equation (9) models the random effects $\beta_{k,t}$ for each covariate $k$ independently for each settlement type $t$ with uninformative priors on the mean $\rho_k$ and variance $\omega_k$ terms, which follow a normal and uniform distribution, respectively.

$$\beta_{k,t} \sim \text{Normal}\left(\rho_k, \omega_k\right) \quad (9)$$

$$\rho_k \sim \text{Normal}\left(0, 1000\right)$$

$$\omega_k \sim \text{Uniform}\left(0, 1000\right)$$

For each covariate $k$, random effects $\beta_{k,t}$ with similar estimated posterior distributions across settlement types $t$ are converted into a fixed effect $\beta_k$ modeled with an uninformative normal distribution (Eq. (10)).

$$\beta_k \sim \text{Normal}\left(0, 1000\right) \quad (10)$$

**Age and sex structure model**. Age and sex structures are modeled as a Dirichlet-multinomial process[21]. This distribution is often used to model compositional

count data, in other words, the count of observations (e.g., people) belonging to mutually exclusive categories (e.g., age and sex groups). Equation (11) models the observed count of people $N_{g,p}$ within an age and sex group $g$ and a province $p$ as a Multinomial process, where $\pi_{g,p}$ is the relative proportion of the age and sex group and $N_p$ the observed population within the province $p$. $g$ comprises $G$ mutually exclusive age and sex groups—two sex groups (i.e., male and female), each subdivided into 18 age groups (i.e., under 1-year-old, 1 to 4 years old, 5-year groups from 5 to 80, and above 80 years old). Age and sex proportions were not modeled at the sub-provincial level because reduced sample sizes could result in spurious estimates for the smallest groups[9,21].

$$N_{g,p} \sim \text{Multinomial}\left(N_p, \pi_{g,p}\right) \quad (11)$$

Because the sum of $\pi_{g,p}$ within each $p$ is constrained to one, Eq. (12) uses an uninformative Dirichlet distribution as a conjugate prior for $\pi_{g,p}$ where $\chi^G$ is a constant numerical vector with values $1/G$ and of length $G$[13].

$$\pi_{g,p} \sim \text{Dirichlet}\left(\chi^G\right) \quad (12)$$

**Model fit and diagnostics**. We estimated the model with MCMC methods in JAGS 4.3.0[39] using the R package runjags[40] version 2.0.4. The convergence of three MCMC chains was assessed using the Gelman-Rubin statistic, and values less than 1.1 were interpreted as indicating convergence[41], while the model residuals were tested for spatial autocorrelation using semivariograms and Moran's I statistics. We examined model fit in- and out-of-sample using 10-fold cross-validation, where the model was fit ten times, each time withholding a random 10% of microcensus clusters until all had been held out once. To assess model fit for age and sex proportions, we held out 10% of the clusters for each province and assessed the combined posterior distribution for each demographic group. For in- and out-of-sample predicted population sizes, densities, and province-level age and sex proportions, we evaluated bias (i.e., the mean of residuals—mean posterior predictions minus observed values), imprecision (i.e., the standard deviation of residuals), inaccuracy (i.e., the mean of absolute residuals), $R^2$ values (i.e., the squared Pearson correlation coefficient among the residuals), and the percentage of observations falling within the 95% prediction intervals. We also computed bias, imprecision, and inaccuracy using standardized residuals (i.e., residuals divided by the mean posterior predictions)[41].

**Reporting summary**. Further information on research design is available in the Nature Research Reporting Summary linked to this article.

## Data availability
The data used in this study are available on Zenodo [https://doi.org/10.5281/zenodo.5712953]. The population data generated in this study can be accessed on the WOPR repository [https://wopr.worldpop.org/?COD/Population/v2.0] and visualized using the WOPR Vision web application [https://apps.worldpop.org/woprVision]. Summary statistics of the posterior distributions generated in this study are available in the Source Data file.

## Code availability
The code developed in this study has been deposited on Zenodo [https://doi.org/10.5281/zenodo.5712953].

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

## Acknowledgements
This work is part of the GRID3 project (Geo-Referenced Infrastructure and Demographic Data for Development), funded by the Bill and Melinda Gates Foundation and the United Kingdom Foreign, Commonwealth and Development Office (FCDO) (#INV009579 A.J.T.). Project partners include WorldPop at the University of Southampton, the United Nations Population Fund (UNFPA), the Center for International Earth Science Information Network (CIESIN) in the Earth Institute at Columbia University, and the Flowminder Foundation. The UCLA-DRC Health Research and Training Program based at the University of California, Los Angeles Fielding School of Public Health, the Kinshasa School of Public Health (KSPH) led the two rounds of household surveys in 2017 and 2018, with the support of the Bureau Central du Recensement (BCR) (#OPP1151786 A.W.R.). Prof Emile Okitolonda-Wemakoy at the KSPH, who passed away before the submission of this work, provided oversight to the household survey data collection. D'Andre Spencer, Camille Dzogang, Jojo Mwanza, Handdy Kalunga, Millet Mfawankang, Eric Musenge, Elie Lokutumba, Joseph Wasiswa, Arthur Lisambo, Kevin Karume, Kizito Mosema, Lievin Dinoka, and Michael Beya supervised the surveyors who were hired locally in collaboration with the provincial health departments. The Oak Ridge National Laboratory (ORNL) supported the first round of household surveys. Key ORNL collaborators include Eric Weber, Jeanette Weaver, and St. Thomas LeDoux. The survey data collection instrument and data quality control platform were developed by eHealth Africa in collaboration with the UCLA-DRC program. Key eHealth Africa collaborators include Ayodele Adeyemo, Dami Sonoiki, and Adeoluwa Akande. The health zone bureau staff throughout the five provinces provided logistical support to surveyors as they traveled to microcensus clusters within each health zone. We acknowledge the work of local surveyors who carried out the survey data collection, often in the face of significant logistical challenges in remote, difficult-to-traverse areas. The authors used the IRIDIS High-Performance Computing Facility and associated support services at the University of Southampton.

## Author contributions
G.B. prepared the manuscript; E.D., D.R.L., C.A.D., H.R.C., A.N.L., K.T., C.S., N.A.H., T.F., K.M., A.B., A.W.R., and A.J.T. edited the manuscript; G.B., E.D., H.R.C., K.T., C.S., N.A.H., T.F., K.M., and A.B. supervised data collection; G.B., E.D, and H.R.C. processed the data; G.B., E.D., D.R.L., and C.A.D. developed the model; G.B., E.D., and D.R.L. implemented the model; H.R.C., A.N.L., K.T., and A.J.T. provided project oversight; A.W.R. and A.J.T. acquired funding.

## Competing interests
The authors declare no competing interests.
