## [Peer Review File · Nature Communications]

Reviewer comments, first round -

Reviewer #1 (Remarks to the Author):

Summary

This study presents a hierarchical Bayesian model to obtain population estimates (and related indicators) from a combination of building footprints and household survey data and applies it to five provinces in the DRC. As such, it responds to an important need for accurate population statistics in policy planning and interventions in developing countries. I have read this paper with great interest and like the method being developed and proposed. However, I think the quality of the paper could be further improved if the following issues were addressed.

Major issues

Despite its usefulness to yield population statistics in setting with imperfect data systems, the paper lacks explicit detail on the major building blocks constituting the statistical method used as well as regarding the interpretation of results. Various data, methods and tools are introduced but remain largely hidden behind references; that is without much description of their content, strengths, limitations and competing alternatives. It would be good and helpful if the reader were a bit more guided throughout the paper. In a similar vein, the discussion of the results could be pushed a bit further to better showcase the added value of the present study.

On household survey data.

- Were the household surveys used to leverage the statistical model particularly conducted for the purpose of this study? Does it comprise the measurement and classification of urban built-up on the ground as well? Was it only conducted in the five provinces mentioned?
- In a similar vein, under what conditions would it be possible to use other household surveys and their enumeration of selected clusters, like in MICS and DHS, to improve the accuracy of the model?

On statistical methods and tools.

- It would be useful to obtain a bit more explanation on the following concepts or tools: (i) Dirichlet-multinomial process (page 7); (ii) the use of the building footprint area as a "multiplicative constraint" (page 10); (iii) how were the 24 municipalities precisely aggregated into 9 contiguous groups, and how does that prevent the presence of unsurveyed sub-provincial units (page 16)?
- The main statistical method (hierarchical Bayesian model) was chosen and applied with little discussion of its strengths and limitations. Why did the authors choose this method over alternative approaches; how does it relate to other bottom-up approaches?
- In a similar vein, the precise extension or added value compared to the initial method as applied in Nigeria is not completely clear to me: has it to do with the integration of a "weighted-precision approach", the addition of "uncertainty measures", or anything else? Simply stated, in what sense, does the current model precisely "advance [...] the state-of-the-art of bottom-up population modeling", as stated in the manuscript?
- In addition to population counts and population densities, the model provides estimations of age/sex breakdowns. I think this aspect could be better explained. More in particular, where does the building footprint data come in to help estimate the age/sex breakdowns? Aren't these simply imputed on the newly estimated population totals, which may explain (partly) why R2 is equal to 100%.

On discussion of results.

- Page 6-7. How should one precisely interpret the positive and negative effects across settlement types for each of the three covariates in the model? Some discussion is provided in the paper, though it remains largely hypothetical with main reference to the presence or absence of non-residential buildings.
- In a similar vein, I have difficulty understanding why the model-of-fit performance for population

densities is worse than for population totals? This question has a technical and interpretational component. Technically, as the model essentially provides estimates of population totals based on building footprint data, I would assume that the same overall fit performance largely applies to both population totals and densities, given that the latter is based on the former. Or, were building footprints also observed/measured on the ground using the household surveys? In terms of interpretation, what is the implication of the large underpredictions in densely populated clusters with reduced coverage of building footprints? Does this mean that people generally live in certain house types that are not well captured by the building footprint data? If so, this could be discussed as a limitation of the data being used.

- In addition, I would like to see how the proposed population estimates compare to other (official) population series, for example those produced by the National Institute of Statistics of the DRC or UNFPA. How much difference do we observe across various population series; and thus, how much inaccuracy is there in current household survey sampling designs and/or policies requiring accurate population statistics?

Other issues

- How are urban and rural settlements precisely distinguished? What criterion has been used? Relatedly, how is the exact spatial extent of a micro cluster being defined/delimited in order to match building footprints with enumerated people using the household survey data?

- There is confusion about the selected provincial capital of Kwilu. The paper refers to Bandundu, but the location on the map (point D in Figure 1) points to Kikwit, which I think is the actual capital of the newly created province after the province of Bandundu was disintegrated into three parts. Moreover, I do not understand why the city of Bandundu (which is no longer a provincial capital, although there might be discussion on this) is considered a rural settlement type, as indicated on Figure 2 located at the most northwestern point of the Kwilu province.

- 21 clusters were discarded because of spurious population densities. It might be useful for the reader to know the origin of this spuriousness: does it relate to the population counts captured by the household survey or to the observations on building footprints.

- On page 5. What are the 37 sub-provincial regions precisely referring to?

- On page 6. What could be the reason of observing a heterogeneous residential context in the urban settlements of the three provinces mentioned?

- Monotone writing style, especially in the section on material and methods. For example, on page 15-16: We used [...]; We summarized [...]; We allocated [...]; We labeled [...]; We accessed [...]; We first derived [...]; We then created [...]; we merged [...].

- Write acronyms in full on first occurrence, such as MCMC on page 8.

Reviewer #2 (Remarks to the Author):

Review for the manuscript 'High-resolution population estimation using household survey data and building footprints'

This study addresses a critical question of how to estimate population and related demographics in areas/countries with outdated data by using a Bayesian hierarchical model integrating population weights and building footprints. It tests out this method in the empirical study in Congo which can be extended to other geographic contexts in less data availability. Overall this article is well written with reasonable scientific soundness and robust modelling techniques. I only have minor queries as below that can be addressed by the authors to improve the overall paper quality.

In the method section of 'building footprints', I am not quite sure how satellite imagery in 2009 and 2019 serve different purposes in the data extraction. Say, satellite imagery in 2019 should be the latest one then is the 2009 data used for calibration?

In 'Covariate processing and selection', the authors indicated that they discarded two factors highly correlated. It would be more beneficial to test out a principal component analysis to retrieve the principal factors rather than deleting colinearly ones which will cause reduced data dimension.

In 'population modeling', the modelling procedure overall looks well but I have an essential question relating to population weights that seems to be retrieved and calculated based on the survey data. I am less sure if the survey data is representative enough since I didn't find any calibration or related information regarding the survey itself. Also, have the authors considered the spatial weights in the modelling? Say, population may tend to concentrate in areas with good service, facilities, and in good locations, which means it may have a distance-decay effect in the population estimates but the current model seems to only consider the population weights at a single/discrete cell (?). It would be better if the authors can better justify this point.

In Table 1, it would be better to clarify what is out-of-sample in the table caption/notes. Also why for age and sex proportions, the R-square is 1? and other measures are all 0? It is quite confusing to see these weird measures in this table as the only one in the manuscript which may cause the doubt from readers about the analytical reliability. I recommend to delete these measures if they are not critical ones or the authors should provide more explanations if they insist to include them here.

Again, overall this paper will benefit the current literature by its analytical framework, modelling techniques and empirical evidence. I recommend a decision of minor revisions before it is further considered for publication.

Reviewer #3 (Remarks to the Author):

The manuscript is very interesting and represents a very original approach in the context of countries with poor statistics like Congo. Since I am not a Bayesian statistician but a demographer, I do not enter into the review of the model and method used even if the basic technique that is presented is well described and I agree with the approach of updating the population counts from the 1984 census on the basis of new settlements (by using building footprints). Since the results obtained partially meet the expectations of the authors (Statistical model residuals still show some levels of uncertainty) and the methodological work behind it seems very significant, it would be better to make it clear from the article that this is an experimental work. The references to the literature are there and only the authors are recalled in the article. The approaches and the results obtained in other countries or within the UNFPA experience are never argued as terms of comparison. In my opinion, it would be necessary to recall them and to highlight the significant aspects of the original results that the authors of the article have reached.

Reviewer 1

Major issues

Despite its usefulness to yield population statistics in setting with imperfect data systems, the paper lacks explicit detail on the major building blocks constituting the statistical method used as well as regarding the interpretation of results. Various data, methods and tools are introduced but remain largely hidden behind references; that is without much description of their content, strengths, limitations and competing alternatives. It would be good and helpful if the reader were a bit more guided throughout the paper. In a similar vein, the discussion of the results could be pushed a bit further to better showcase the added value of the present study.

We thank Reviewer 1 for the insightful feedback. We have thoroughly revised the Methods and the Results sections of the manuscript as well as the Discussion section. We address the comments and requests of Reviewer 1 in detail below.

On household survey data

- Were the household surveys used to leverage the statistical model particularly conducted for the purpose of this study? Does it comprise the measurement and classification of urban built-up on the ground as well? Was it only conducted in the five provinces mentioned?

The two rounds of household surveys were carried out for the purpose of this study only in the five provinces located in the Western part of the DRC. Only essential demographic data (e.g., age and sex) were collected in the household surveys because of the time and resources necessary to fully enumerate the 926 clusters. We have revised the first paragraph of the Household surveys subsection (Methods section) to describe the household survey data more in detail.

- In a similar vein, under what conditions would it be possible to use other household surveys and their enumeration of selected clusters, like in MICS and DHS, to improve the accuracy of the model?

The weighted-precision approach developed in this study tackles one of the main obstacles to incorporating data from different household surveys into bottom-up population models, namely the probabilistic selection of clusters. Another essential condition is the availability of information on the geographic distribution of the population of the cluster (i.e., the cluster boundary or the GPS location of every household). When this condition is met, bottom-up population models can be flexibly customized to overcome other potential constraints (e.g., data collected in different periods). We have substantially extended the sixth paragraph of the Discussion section to explain these essential conditions.

1.2. On statistical methods and tools

- It would be useful to obtain a bit more explanation on the following concepts or tools: (i) Dirichlet-multinomial process (page 7); (ii) the use of the building footprint area as a “multiplicative constraint” (page 10); (iii) how were the 24 municipalities precisely aggregated into 9 contiguous groups, and how does that prevent the presence of unsurveyed sub-provincial units (page 16)?

We have thoroughly revised the manuscript to ensure that the statistical methods and tools are sufficiently described. In particular, (i) we have extended the Age and sex structure model subsection (Methods section) to discuss key characteristics and applications of the Dirichlet-multinomial distribution; (ii) we acknowledge that, in its original position, the sentence was confusing and have moved it to the third paragraph of the Discussion section; (iii) we merged the boundaries of the 24 municipalities comprising the city of Kinshasa into nine groups with similar settlement characteristics to ensure that each group contained at least one microcensus cluster. This is because the population densities estimated within each group informed the random intercepts in the population model. The grouping was defined using information available in the Strategic Orientation Plan for the Agglomeration of Kinshasa. We have extended the Administrative boundaries subsection (Methods section) to clarify the assumptions and processing steps involved in the grouping of municipalities.

- The main statistical method (hierarchical Bayesian model) was chosen and applied with little discussion of its strengths and limitations. Why did the authors choose this method over alternative approaches; how does it relate to other bottom-up approaches?

We have revised the first paragraph of the Introduction section by contrasting projection models, the most common method for population estimation, and bottom-up models. We have also revised the first paragraph of the Population model subsection (Methods section) and the last paragraph of the Discussion section to introduce the strengths and limitations of hierarchical Bayesian models for bottom-up population modeling.

- In a similar vein, the precise extension or added value compared to the initial method as applied in Nigeria is not completely clear to me: has it to do with the integration of a “weighted-precision approach”, the addition of “uncertainty measures”, or anything else? Simply stated, in what sense, does the current model precisely “advance [...] the state-of-the-art of bottom-up population modeling”, as stated in the manuscript?

In this work, we extend the framework developed for bottom-up population modeling in Nigeria to leverage two new types of input data available in our study region: comprehensive household surveys with a complex sampling design and building footprints. In doing so, we introduced the following innovative model components — (i) weighted-precision to account for probabilistic sampling designs, (ii) Dirichlet-multinomial estimation of age and sex structures, and (iii) inclusion of key attributes derived from building footprints in the model. We have revised the last paragraph of the Introduction section and the sixth paragraph of the Discussion section to better describe the added value of our study.

- In addition to population counts and population densities, the model provides estimations of age/sex breakdowns. I think this aspect could be better explained. More in particular, where does the building footprint data come in to help estimate the age/sex breakdowns? Aren't these simply imputed on the newly estimated population totals, which may explain (partly) why R2 is equal to 100%.

We have expanded the Age and sex structure model subsection (Methods section) to clarify the use of the Dirichlet-multinomial model in the estimation of age and sex breakdowns. The use of building footprints is limited to the estimation of population densities and totals, which are then disaggregated using the age and sex proportions estimated using the Dirichlet-multinomial model. Given that proportions are estimated at the province level, an R2 value of 1.00 appears to be an indication of the limited variability in the age and sex data at the province level, which can be fully captured by the model. However, these results should be interpreted with caution because small differences in the estimated proportions (e.g. 0.001) can have a substantial impact on the total population allocated to each age and sex class in a province. We have added a sentence providing an interpretation of the R2 value in the Model diagnostics subsection and another sentence at the end of the second paragraph of the Discussion section to contextualize these results.

On the discussion of results

- Page 6-7. How should one precisely interpret the positive and negative effects across settlement types for each of the three covariates in the model? Some discussion is provided in the paper, though it remains largely hypothetical with main reference to the presence or absence of non-residential buildings.

Given that our hierarchical model was not designed for causal inference, we argue that it would be misleading to quantify and interpret the covariate effects. As pointed out by Reviewer 1, we described the direction of the associations in the Covariate effects

subsection (Results section) and commented on the potential reasons for these associations in the Discussion section. However, the model covariates are topological and morphological attributes derived from building footprints that offer limited practical interpretability in the context of our study. For instance, average building proximity in rural settlements has a mean covariate effect of 0.15. This indicates that the increase of one unit in the covariate value results in an average increase in the mean population density of 1.16, that is $\exp(0.15)$, within rural settlements. We have added a sentence at the end of the fourth paragraph of the Discussion section mentioning that the covariates were not selected for causal inference and, for this reason, the interpretation of their effects is hypothetical.

- In a similar vein, I have difficulty understanding why the model-of-fit performance for population densities is worse than for population totals? This question has a technical and interpretational component. Technically, as the model essentially provides estimates of population totals based on building footprint data, I would assume that the same overall fit performance largely applies to both population totals and densities, given that the latter is based on the former. Or, were building footprints also observed/measured on the ground using the household surveys? In terms of interpretation, what is the implication of the large underpredictions in densely populated clusters with reduced coverage of building footprints? Does this mean that people generally live in certain house types that are not well captured by the building footprint data? If so, this could be discussed as a limitation of the data being used.

In this study, we only used building footprints automatically extracted by Ecopia.AI using satellite imagery provided by Maxar Technologies. In the Population model subsection (Methods section), we define our Bayesian hierarchical model in two subsequent steps. The first step models population densities as a lognormal process (Eq. 2), while the second models population totals as a Poisson process, where the rate parameter is defined through the multiplication of the estimated population densities by the observed building footprint area (Eq. 1). This second step leverages an additional source of information, namely the observed building footprint area, which acts as a multiplicative constraint in the estimation of population totals. For instance, when estimating population totals in scarcely populated clusters (e.g. 10 people), the impact of important variations in the estimated population densities (e.g., 1,000 or 2,000 people/ building footprint ha) is mitigated by the small magnitude of the variability in the multiplication factor (e.g., 0.001 or 0.005 observed building footprint ha). Figure 6 confirms this interpretation, where substantial underpredictions of population density in a number of rural clusters were not observed for population totals, thus explaining the different goodness-of-fit between the population totals and densities. The clusters exhibiting substantial underpredictions of population densities appear to be characterized by inflated observed population densities because the number of building footprints used to derive the areal denominator (i.e. people/built-up ha) was significantly lower than the number of surveyed buildings. The reason for this could be found in the satellite imagery used to delineate building footprints that, in these clusters, were partly

obfuscated, thus preventing the detection of buildings. Given that our study aimed at predicting population totals, a reduced goodness-of-fit for population densities was not seen as an issue. We have added a sentence at the end of the first paragraph of the Population model subsection (Methods section) to stress the benefits of using building footprints in our model, and expanded the last paragraph of the Model diagnostics subsection (Results section) and third paragraph of the Discussion section to better describe and interpret the reasons for a reduced model fit for population densities.

- In addition, I would like to see how the proposed population estimates compare to other (official) population series, for example those produced by the National Institute of Statistics of the DRC or UNFPA. How much difference do we observe across various population series; and thus, how much inaccuracy is there in current household survey sampling designs and/or policies requiring accurate population statistics?

In the DRC context, it is difficult to produce a meaningful comparison of any sort of population estimate because the last national population and housing census was carried out in 1984, and no other systematic enumeration of the Congolese population has been carried out since. This issue is well summarized by Marivoet and De Herdt (2017), which state that “nobody really seems to know how many Congolese today populate the DRC.” In addition, official figures provided by the Congolese Institut National de la Statistique (INS) or international bodies such as the United Nations Population Division (UNPD) are the result of projection models based on the “outdated and incompletely processed census of 1984” (Marivoet and De Herdt, 2017), with specific assumptions and limitations that make it hard to compare with the bottom-up estimates produced in this study. While projections models developed by international bodies typically fail to provide population estimates at the subnational level, the projection models developed by the INS estimates population sizes at the provincial level, potentially enabling a comparison with our aggregated estimates. However, the lack of transparency on the input data and methods adopted by the INS discouraged us from developing any comparison because of the many variables that could invalidate such exercise. For this reason, we argue that in the absence of a recent national census our estimates should be compared with the enumeration of small areas that are regularly carried out as part of public health campaigns. A good example is provided by the SAPIENS project (<https://schistosomiasiscontrolinitiative.org/sapiens-project>), which aims at comparing the population estimates produced in this study against other population data. We have added a sentence at the end of the Population estimates subsection (Results section) and substantially revised the sixth paragraph of the Discussion section to explain why it would not be informative to compare our population estimates with alternative projection-based estimates.

Other issues

- How are urban and rural settlements precisely distinguished? What criterion has been used? Relatedly, how is the exact spatial extent of a micro cluster being defined/delimited

in order to match building footprints with enumerated people using the household survey data?

Urban and rural settlements have been defined using classes originally defined by the Center for International Earth Science Information Network (CIESIN) solely from the building footprint data (<https://doi.org/10.7916/d8-cpry-wv37>). The advantage of this approach is that it provides a consistent methodology to define settlement classes. Urban settlements, originally named Built-Up areas (BUAs), are characterized by contours with an area greater than or equal to 400,000 square meters that maintains a building density of thirteen or more across the entire area. Rural settlements are a combination of the original Small Settlements (SSAs) and Hamlets classes, which represent all the remaining settlements. We have revised the Building footprints subsection (Methods section) to provide this additional information. The delineation of microcensus clusters was carried out manually by creating polygons including approximately three hectares of settled area and with similar characteristics assessed from satellite imagery. Local surveyors then visited each cluster and fully enumerated each household. We have expanded the second paragraph of the Household surveys subsection (Methods section) to clarify this point.

- There is confusion about the selected provincial capital of Kwilu. The paper refers to Bandundu, but the location on the map (point D in Figure 1) points to Kikwit, which I think is the actual capital of the newly created province after the province of Bandundu was disintegrated into three parts. Moreover, I do not understand why the city of Bandundu (which is no longer a provincial capital, although there might be discussion on this) is considered a rural settlement type, as indicated on Figure 2 located at the most northwestern point of the Kwilu province.

According to official sources of information, including the Journal Officiel de la Republique Democratique du Congo (at page 21) (<http://www.leganet.cd/Legislation/JO/2015/Numeros/JOS.28.03.2015.pdf>), the city of Bandundu is the capital of the Kwilu province. However, in Figure 1, we wrongly centered the inset map D on the city of Bulungu. We have revised Figure 1 to fix this error. Figure 2 shows the spatial distribution of the surveyed clusters. In the Kwilu province, the clusters were selected using spatial random sampling, and only a rural area on the outskirts of the city of Bandundu was surveyed.

- 21 clusters were discarded because of spurious population densities. It might be useful for the reader to know the origin of this spuriousness: does it relate to the population counts captured by the household survey or to the observations on building footprints.

We apologize for the confusion. In these 21 clusters, there were spurious densities for two reasons: in 7 clusters we observed reduced survey coverage due to inaccessible areas (resulting in undercounting of people) and in 14 clusters we observed no building footprints

coverage (resulting in undercounting of building footprints). We have rephrased the sentence.

- On page 5. What are the 37 sub-provincial regions precisely referring to?

The sub-provincial regions consist of territories, cities, and groups of municipalities for the city-province of Kinshasa defined in the Administrative boundaries subsection (Methods section). We have revised this sentence accordingly.

- On page 6. What could be the reason of observing an heterogenous residential context in the urban settlements of the three provinces mentioned?

Figure 3 shows that model uncertainty (i.e., the 95% credible intervals) is larger in urban settlements, particularly in the provinces of Kongo Central, Kwango, and Mai-Ndombe. This suggests that, in these urban settlements, the evidence provided by the data suggests a greater variability in the average population densities (i.e., model intercepts). We assume that contrasting evidence in the input data in these regions is linked to a highly heterogeneous residential context, ranging from low-density settlements to high-density urban centers. We have revised this sentence to describe the heterogeneous residential context in these provinces.

Monotone writing style, especially in the section on material and methods. For example, on page 15-16: We used [...]; We summarized [...]; We allocated [...]; We labeled [...]; We accessed [...]; We first derived [...]; We then created [...]; we merged [...].

We have thoroughly revised the manuscript to improve its style.

-- Write acronyms in full on first occurrence, such as MCMC on page 8.

We have defined MCMC in full at the first occurrence.

Reviewer 2

This study addresses a critical question of how to estimate population and related demographics in areas/countries with outdated data by using a Bayesian hierarchical model integrating population weights and building footprints. It tests out this method in the empirical study in Congo which can be expanded to other geographic contexts in less data availability. Overall this article is well written with reasonable scientific soundness and robust modelling techniques. I only have minor queries as below that can be addressed by the authors to improve the overall paper quality.

In the method section of 'building footprints', I am not quite sure how satellite imagery in 2009 and 2019 serve different purposes in the data extraction. Say, satellite imagery in 2019 should be the latest one then is the 2009 data used for calibration?

We thank Reviewer 2 for the constructive feedback and for raising this important question. The imagery used for feature extraction was selected to provide the best quality and the most recent representation of man-made structures visible on the ground. For instance, in a region where satellite images taken in 2019 were substantially obscured by clouds, older images were used for the automatic extraction of building footprints. We have substantially revised and expanded the Building footprint subsection (Methods section) to better explain the characteristics of the building footprint data used in this study.

In 'Covariate processing and selection', the authors indicated that they discarded two factors highly correlated. It would be more beneficial to test out a principal component analysis to retrieve the principal factors rather than deleting colinearly ones which will cause reduced data dimension.

We first selected covariates with a strong linear correlation to log-population densities and then discarded the ones that were highly correlated among them to prevent multicollinearity. This procedure enabled us to detect a strong correlation between the covariate "Average Building Area" and Average Building Perimeter." We removed the latter because it had a slightly weaker correlation to log-population densities. Indeed, we could have developed a PCA to retrieve the principal components from the two covariates and use them as covariates in the model. However, we decided not to carry out a dimensionality reduction because the two covariates appeared redundant and the potential improvement in the goodness-of-fit of the model would have most likely been minimal. We have revised the last part of the Covariate processing and selection subsection (Methods section) to improve the description of the covariate selection process.

In 'population modelling', the modelling procedure overall looks well but I have an essential question relating to population weights that seems to be retrieved and calculated based on the survey data. I am less sure if the survey data is representative enough since I didn't find any calibration or related information regarding the survey itself. Also, have the authors considered the spatial weights in the modelling? Say, population may tend to concentrate in areas with good service, facilities, and in good locations, which means it may have a distance-decay effect in the population estimates but the current model seems to only consider the population weights at a single/discrete cell (?). It would be better if the authors can better justify this point.

While in the first round of household surveys, clusters were randomly selected within each stratum, in the second round, they were drawn using a stratified population-weighted sampling design described in Boo et al. 2020 (<https://gatesopenresearch.org/articles/4-13>). For this reason, we retrieved the sampling weights for each cluster and used this information in the model to produce unbiased estimates of population densities. The complex sampling design was meant to capture as best as possible the spatial distribution of population densities and demographics across the five provinces by incorporating contextual strata approximating the urban-rural divide. We have uploaded data and code to replicate this

study on Zenodo (<https://doi.org/10.5281/zenodo.5712953>). Because of data privacy and confidentiality, the survey data and protocol are not publicly available. However, we would be able to disclose individual aspects of the protocol upon request. The inclusion of spatial weights is necessary when the model fails to capture an underlying spatial process, a condition highlighted by spatial autocorrelation in the model residuals. We carried out an analysis of the model residuals using the Moran's I test and no significant spatial autocorrelation was detected. For this reason, we assume that the model structure and covariates adequately accounted for the spatial processes associated with population densities. We have added a sentence at the end of the Results section mentioning that no significant spatial autocorrelation in the model residuals was detected using Moran's I test. In addition, we have tested model covariates associated with distance to services (e.g., shops, hospitals, schools, etc.) derived from OpenStreetMap data but none of these was strongly correlated with population densities. We argue that the limited completeness of OSM data in the region may be one of the causes of this lack of correlation. Unfortunately, more comprehensive data on the location of services was not available to us at the time of the study.

In Table 1, it would be better to clarify what is out-of-sample in the table caption/notes. Also why for age and sex proportions, the R-square is 1? and other measures are all 0? It is quite confusing to see these weird measure in this table as the only one in the manuscript which may cause the doubt from readers about the analytical reliability. I recommend to delete these measures if they are not critical ones or the authors should provide more explanations if they insist to include them here.

We have added a definition of out-of-sample in the caption of Table 1. An R^2 value of 1.00 for age and sex proportions indicates that the model captures the variability in the observed age and sex proportions at province level. A value of 0.00 in terms of bias, imprecision, and accuracy is another crucial piece of information confirming the goodness-of-fit of the model. We contend that these measures are informative about the process being modeled and decided to keep them. When interpreting these results, it is important to consider that the variability in the proportions is limited because they can span between 0.00 and 1.00, and this variability is further limited by the relatively large number of age and sex classes ($n=36$). However, a slight difference in the estimated proportions (e.g., 0.001) can have a large impact (e.g., thousands of people) on the population allocated to each class in a province. For this reason, in future studies, we will attempt to model age and sex proportions at the sub-provincial level, where variability in the data is expected to be larger. We have substantially extended the Age and sex structure model subsection (Methods section), added a sentence in the first paragraph of the Model diagnostics subsection (Methods section), and revised part of the second and the fifth paragraph of the Discussion section to better contextualize these results.

Reviewer 3

The manuscript is very interesting and represents a very original approach in the context of countries with poor statistics like Congo. Since I am not a Bayesian statistician but a demographer, I do not enter into the review of the model and method used even if the basic technique that is presented is well described and I agree with the approach of updating the population counts from the 1984 census on the basis of new settlements (by using building footprints).

Since the results obtained partially meet the expectations of the authors (Statistical model residuals still show some levels of uncertainty) and the methodological work behind it seems very significant, it would be better to make it clear from the article that this is an experimental work.

We thank Reviewer 3 for the overall positive feedback and the insightful remarks. Indeed, as also pointed out by Reviewer 1, our modeling effort is particularly valuable in setting with imperfect data systems. This observation is also confirmed by UNFPA that, in a recent technical note (<https://www.unfpa.org/resources/value-modelled-population-estimates-census-planning-and-preparation>), highlighted the role of bottom-up population models for census planning and preparation. Indeed, a statistical model is by design an approximation of reality with some degree of uncertainty. In our model, uncertainty is assessed using credibility intervals based on Bayesian posterior probabilities, a useful measure to support effective decision-making and planning. We have added a sentence at the end of the third paragraph of the Introduction section to showcase the advantages of working with uncertainty in a Bayesian context and revised the first and last paragraph of the Discussion section to describe the applicability and limits of our model. This approach to population estimation is not seen as experimental because the results of similar bottom-up models developed in Zambia and Burkina Faso (but not yet published in peer-reviewed journals) have been adopted by the respective National Statistical Offices.

The approaches and the results obtained in other countries or within the UNFPA experience are never argued as terms of comparison. In my opinion, it would be necessary to recall them and to highlight the significant aspects of the original results that the authors of the article have reached.

We thank Reviewer 3 for raising this essential point. At the time of the study, bottom-up population estimates were produced only in a handful of countries (<https://wopr.worldpop.org/?/Population>) and only the Nigeria model developed by Leasure et al. (2020) was published in a peer-reviewed journal. We extensively refer to the Nigeria model throughout the manuscript. As also mentioned in our response to Reviewer 1, it is difficult to produce a meaningful comparison of any sort of population estimate in the DRC because the last national population and housing census was carried out in 1984, and no other systematic enumeration of the Congolese population has been carried out since. In addition, official estimates provided by the Congolese Institut National de la Statistique (INS)

or international bodies such as UNPD are the result of projection models based on the “outdated and incompletely processed census of 1984” (Marivoet and De Herdt, 2017), with specific assumptions and limitations that make it hard to compare with the bottom-up estimates produced in this study. For this reason, we argue that our estimates should be compared with the enumeration of small areas that are frequently carried out as part of public health campaigns. A good example is provided by the SAPIENS project (<https://schistosomiasiscontrolinitiative.org/sapiens-project>), which aims at comparing the population estimates produced in this study versus official population data. Following Reviewer 1 and Reviewer 3’s requests, we have added a sentence at the end of the Population estimates subsection (Results section) and substantially revised the sixth paragraph of the Discussion section to explain why it would not be informative to compare our population estimates with alternative projection-based estimates and reference the SAPIENS project as an excellent initiative to validate our estimates.

In conclusion, we would like to thank you and the reviewers for providing constructive criticism and making valuable suggestions. We believe that your comments have helped us to improve our study. In addressing the comments above and in the original manuscript, we hope that this is now to your satisfaction.

Sincerely,

Dr. Gianluca Boo and co-authors

Reviewer comments, second round -

Reviewer #1 (Remarks to the Author):

This is an interesting, methodologically sound and useful study, which I would recommend publishing in Nature Communications. The authors have provided detailed and valuable feedback to the various issues raised and have reworked different parts of the text accordingly.

Despite being computable from the WorldPop Open Population Repository (WOPR), I still find it a bit unfortunate however that no inferences were drawn with other population estimates, let alone that they are made explicit. For example, how many people live in Kinshasa today according to this Bayesian hierarchical model? As mentioned in the text, this question is key for development policy & planning; yet it does not receive an explicit answer.

Reviewer #2 (Remarks to the Author):

Thanks for the author(s)' efforts on improving the manuscript and addressed the concerns I raised in the previous review. I don't have any further questions and happy to support this manuscript to be considered for a formal publication.

Reviewer #3 (Remarks to the Author):

The manuscript seems to me to have improved greatly in text quality and overall layout. The suggestions that I had asked to be made have been accepted, and I note that on the whole the manuscript can be read very well and is very dense with scientific information. I also like the figures very much because they are well done both in form and in the use of colors. Overall I think it is a very good work and I do not have any further suggestions to propose.

15 February 2022

NCOMMS-21-22940A — Manuscript Revision

Reviewer 1

Remarks to the Author

This is an interesting, methodologically sound and useful study, which I would recommend publishing in Nature Communications. The authors have provided detailed and valuable feedback to the various issues raised and have reworked different parts of the text accordingly.

Despite being computable from the WorldPop Open Population Repository (WOPR), I still find it a bit unfortunate however that no inferences were drawn with other population estimates, let alone that they are made explicit. For example, how many people live in Kinshasa today according to this Bayesian hierarchical model? As mentioned in the text, this question is key for development policy & planning; yet it does not receive an explicit answer.

We thank Reviewer 1 for the careful and insightful review of our manuscript and the final comments provided.

Indeed, the population estimates produced in this study can be used to compute the number of people within administrative or other areal units. However, these figures can be very different according to the boundaries used for the spatial aggregation of the grid-cell-level population estimates. For instance, existing population projections for Kinshasa refer more or less explicitly to the urban agglomeration, the municipality, or the province. Furthermore, population figures for the same unit often imply different spatial extents because the units' boundaries change according to the data provider. Given this additional source of uncertainty around the enumeration of the Congolese population, we decided not to include any aggregated estimates in this manuscript. Still, we strongly advocate for the use of spatially aggregated population estimates within user-defined boundaries and further comparisons based on the same spatial units using WOPR Vision on <https://apps.worldpop.org/woprVision>.

We have completed the sixth paragraph of the Discussion section to describe this additional source of uncertainty in the comparison between bottom-up estimates and other population figures.

Reviewer 2

Remarks to the Author

Thanks for the author(s)' efforts on improving the manuscript and addressed the concerns I raised in the previous review. I don't have any further questions and happy to support this manuscript to be considered for a formal publication.

We are grateful to Reviewer 2 for the thorough review of our manuscript and the constructive feedback provided throughout the revision process.

Reviewer 3

Remarks to the Author

The manuscript seems to me to have improved greatly in text quality and overall layout. The suggestions that I had asked to be made have been accepted, and I note that on the whole the manuscript can be read very well and is very dense with scientific information. I also like the figures very much because they are well done both in form and in the use of colors.

Overall I think it is a very good work and I do not have any further suggestions to propose.

We are pleased that Reviewer 3 is satisfied with the current version of our manuscript and thank them for the constructive criticism and overall positive feedback.

We renew our thanks to the reviewers for providing constructive criticism and making valuable suggestions. We believe that their comments have helped us to improve our manuscript and hope that this is now to their satisfaction.

Sincerely,

Dr. Gianluca Boo and co-authors